# Research Advances in Clinical Applications, Anticancer Mechanism, Total Chemical Synthesis, Semi-Synthesis and Biosynthesis of Paclitaxel

**DOI:** 10.3390/molecules28227517

**Published:** 2023-11-10

**Authors:** Shengnan Zhang, Taiqiang Ye, Yibin Liu, Guige Hou, Qibao Wang, Fenglan Zhao, Feng Li, Qingguo Meng

**Affiliations:** 1Key Laboratory of Molecular Pharmacology and Drug Evaluation (Ministry of Education), Collaborative Innovation Center of Advanced Drug Delivery System and Biotech Drugs in Universities of Shandong, School of Pharmacy, Yantai University, Yantai 264005, China; shengnan0407@163.com (S.Z.); yetaiqiang@s.ytu.edu.cn (T.Y.); yibinliu98@163.com (Y.L.); zfl123@ytu.edu.cn (F.Z.); 2School of Pharmacy, Binzhou Medical University, Yantai 264003, China; guigehou@163.com; 3School of Biological Science, Jining Medical University, Rizhao 276800, China; qibaowang@mail.jnmc.edu.cn

**Keywords:** paclitaxel, anticancer mechanism, total synthesis, semi-synthesis, biosynthesis

## Abstract

Paclitaxel, a natural secondary metabolite isolated and purified from the bark of the *Taxus* tree, is considered one of the most successful natural anticancer drugs due to its low toxicity, high potency and broad-spectrum anticancer activity. *Taxus* trees are scarce and slow-growing, and with extremely low paclitaxel content, the contradiction between supply and demand in the market is becoming more and more intense. Therefore, researchers have tried to obtain paclitaxel by various methods such as chemical synthesis, artificial culture, microbial fermentation and tissue cell culture to meet the clinical demand for this drug. This paper provides a comprehensive overview of paclitaxel extraction, combination therapy, total synthesis, semi-synthesis and biosynthesis in recent years and provides an outlook, aiming to provide a theoretical basis and reference for further research on the production and application of paclitaxel in the future.

## 1. Introduction

Nature has been the source of medicinal products for thousands of years, with many drugs derived from plants. As a model for drug discovery from natural products, paclitaxel (registered as Taxol^®^ by BMS (New York, NY, USA)) is one of the most successful anticancer drugs of the past 50 years. The global paclitaxel market was valued at US$4.51 billion in 2021 and is expected to reach over US$11.16 billion by 2030 [1]. Paclitaxel was first isolated and purified from the bark of *Taxus brevifolia*, which is a rare and slow-growing evergreen found in the old-growth forests of the Pacific Northwest (also known as the yew tree), and its structure was characterized by Wani et al. in 1971 [2]. Paclitaxel possesses a highly oxygenated tetracyclic skeleton with a bridged bicyclo [5.3.1] undecane ring system (Figure 1). The anti-tumor activity of paclitaxel is mainly due to the C13 side chain, A ring, oxetane ring and C2 benzoyl group [3]. Paclitaxel exists in the form of a white crystalline powder that is highly lipophilic and thus very insoluble in water.

The anticancer activity of paclitaxel was demonstrated in the mouse melanoma B16 model in 1976 [4]. Subsequently, Horwitz et al. found that paclitaxel inhibited cancer cell proliferation by stabilizing microtubules, especially in melanoma cells and ovarian cancer cells [5]. Paclitaxel was initially approved by the U.S. Food and Drug Administration (FDA) in 1992 for the treatment of ovarian cancer and in 1994 for the treatment of metastatic breast cancer [6]. In subsequent years, it has also been approved to treat non-small cell lung carcinoma, AIDS-related Kaposi’s sarcoma and cancers of the lung, bladder, esophagus, prostate and pancreas, either alone or in combination with other anticancer drugs [7,8]. It has been clinically proven that paclitaxel has good anti-tumor effects, especially for ovarian cancer, uterine cancer and breast cancer, which have a high incidence of occurrence [9].

Acquiring adequate supply has been a major challenge throughout the development of paclitaxel. Economically synthesizing paclitaxel is very complex, and isolating it from natural sources is cumbersome. According to a report from CEC China Pharmaceuticals Ltd. (Shanghai, China), 10,000 kg of leaves and bark from *Taxus chinensis* are required to isolate 1 kg of paclitaxel. Even in the most productive species, *Taxus brevifolia*, the paclitaxel content is only 0.01–0.05% [10]. Therefore, direct extraction methods cannot support the large-scale production of paclitaxel and have raised considerable environmental concerns. With ongoing efforts dedicated to paclitaxel production, at least two promising approaches have been developed to address supply and ecological challenges. The first large-scale approach involves the semi-synthesis of paclitaxel. The method was derived to extract 10-deacetylbaccatin III (10-DAB) or baccatin III from renewable twigs of *Taxus* species, which is subsequently converted to paclitaxel [11]. Another method for large-scale production of paclitaxel is cell fermentation using *Taxus* cell suspension cultures. At the moment, Python Biotech is the largest producer of paclitaxel by this method [1]. Although many advances have been made in the production of paclitaxel over the years, there are still several drawbacks to the current approaches, and the problems of supply shortages and high costs of production still cannot be ignored.

In recent years, synthetic biology technology has brought a green and sustainable strategy for the large-scale production of structurally complex and rare natural products through artificially building and optimizing biosynthetic pathways of target compounds in microbial chassis cells. Therefore, analyzing the biosynthetic pathway of paclitaxel and constructing this biosynthetic pathway in microorganisms using synthetic biology methods may be a new route to overcome the paclitaxel supply problem. Despite some progress in this field [12], there are still gaps in our understanding of paclitaxel biosynthesis pathway and its regulatory mechanisms that hinder paclitaxel production using biology methods. Our goal in writing this review was to provide a comprehensive review of paclitaxel extraction, total synthesis, semi-synthesis and biosynthesis methods. We have emphasized the evolution of each approach and highlighted the merits and demerits of each. Ultimately, we hope to provide a theoretical basis and reference for further research on the production and utilization of paclitaxel.

## 2. Anticancer Mechanism and Clinical Applications

Unlike traditional anticancer drugs, paclitaxel neither affects the synthesis of DNA and RNA in cancer cells nor damages DNA molecules, and its mechanism of action is mainly to promote the polymerization of tubulin [13,14]. Tubulin, the basic structural unit of intracellular microtubules, is a heterodimer formed by the polymerization of α-tubulin and β-tubulin molecules. Paclitaxel selectively binds to β-tubulin and promotes the polymerization and assembly of tubulin, which depletes intracellular tubulin, prevents spindle formation, leads to mitotic arrest in G2/M phase, terminates cell division and ultimately leads to cancer cell death (Figure 2) [15].

Paclitaxel also induces the expression of genes and cytokines that inhibit tumor cell growth and apoptosis. Paclitaxel inhibits regulatory cells (Tregs) and tumor-associated macrophages (TAMs), stimulates anti-tumor immunity and leads to the release of pro-apoptotic molecules such as Fas L, TNF-related apoptosis-inducing ligand (TRAIL) and cytokines such as TNF-α and IFN-γ [16,17]. It can also induce apoptosis by activating signaling pathways. Paclitaxel activates phosphatase and tensin homologs deleted on chromosome ten (PTEN) and inhibits PI3K/Akt expression and phosphorylation by generating an excess of ROS and promoting miR-22 overexpression [18]. Another study showed that paclitaxel could upregulate miR-145 and directly inhibit the expression of Akt, thereby inducing apoptosis [19,20].

Paclitaxel may inhibit tumor cell growth by inducing autophagy, but this effect is dependent on the type of cells and the concentration of paclitaxel. One study found that the treatment of MDA-MB-231 cells with paclitaxel (24 μM) induced autophagy but showed no significant inhibition, and combined treatment of cancer cells with paclitaxel and Bridgestone induced the significant inhibition of cancer cells [21]. Other anti-tumor mechanisms of paclitaxel are cellular pyroptosis, senescence and ferroptosis [22,23,24,25,26,27].

The main paclitaxel-like compounds currently in clinical use are paclitaxel, docetaxel (registered as Taxotere^®^ by Sanofi-Aventis (Paris, France)) and cabazitaxel (registered as JEVTANA^®^ by Sanofi-Aventis) (Figure 3) [28]. Docetaxel is structurally similar to paclitaxel, with the difference being that the C10 position is a hydroxyl group instead of an acetyl group and the C3′ position is a Boc group instead of a benzoyl group. In 1996, docetaxel was marketed for the treatment of breast cancer, colon cancer and NSCLC [29]. It binds microtubulin better than paclitaxel and exhibits better solubility, bioavailability and anti-tumor activity. Cabazitaxel, another paclitaxel-based anti-tumor agent, was approved by the FDA in 2010 for the treatment of advanced prostate cancer and can be used in combination with prednisone to treat hormone-refractory metastatic prostate cancer [30,31]. In addition, the new drug larotaxel has completed phase III clinical evaluation for breast cancer alone, and the ternary ring in its structure is thought to minimize P-glycoprotein recognition, potentially overcoming multidrug resistance mechanisms and crossing the blood–brain barrier [32]. Conmotaxel is an access to the paclitaxel structure to inhibit NOD2-mediated inflammatory signaling pathway, which can enhance the therapeutic effect of paclitaxel and inhibit tumor metastasis and has received clinical approval [33] (Figure 3).

Paclitaxel-based antineoplastic drugs are mostly used as first-line anticancer drugs, often in combination with other anticancer drugs (Table 1 and Table 2). Numerous clinical evaluations have found that the combination of paclitaxel and platinum-based drugs for the treatment of advanced solid tumors has the advantages of high survival rate and good tolerance, safety and efficacy for esophageal, ovarian epithelial, cervical and gastric cancers [34,35,36]. In addition to its combination with chemotherapeutic drugs, paclitaxel is often used as a radiotherapy sensitizer in the treatment of squamous cell carcinomas such as intermediate-to-advanced head and neck cancer and nasopharyngeal carcinoma, and it participates in radiotherapy to minimize the duration of treatment with acceptable tolerability and good local control [37,38].

In recent years, paclitaxel has also also often used in combination with other natural drugs, as in the case of co-administration with resveratrol against hepatocellular carcinoma, laryngeal carcinoma and gastric carcinoma, which can improve the anti-tumor activity of the drugs used alone and reduce the dosage and side effects of the two drugs alone [39]. It was shown that paclitaxel elevated the expression of caspase-3, caspase-8, Bax (Bcl-2 assaciated X protein), p53, Fas (factor associated suicide), Fas L (factor associated suicide ligand), cIAP-2 (cIap, cellular inhibitor of apoptosis), NF-кB and epidermal growth factor receptor (EGFR) mRNAs and proteins in HepG2 human liver cancer cells, and resveratrol enhanced the changes in the expression of these mRNAs [40]. The combination of paclitaxel and curcumin reversed multidrug resistance of paclitaxel and inhibited cancer cell growth [41,42]. Furthermore, combination therapy with the two improved the anti-glioma efficacy and helped reduce the side effects of cytotoxic treatment [43]. Curcumin can enhance the anticancer effect of paclitaxel in ovarian cancer by modulating the miR-9-5p/BRCA1 axis [44]. When used in combination for the treatment of lung cancer, curcumin enhanced the growth inhibition of lung cancer H1299 cells and showed a significantly lower IC_50_ value than that of paclitaxel alone [45].

## 3. Sources and Production Methods of Paclitaxel

### 3.1. Extraction from Taxus Plants

Direct extraction of paclitaxel from *Taxus* plants has always been the main method for paclitaxel preparation. However, it is not realistic to supply paclitaxel by extraction from wild natural resources due to its extremely low concentration and the slow growth of yew trees. To meet the increased demand for clinical use and to preserve the wild *Taxus* species, artificial cultivation has been utilized to alleviate the shortage of paclitaxel. For example, two seedling bases were established in Sichuan Province, China, including the Bei-chuan and Hong-ya bases. The yew seedlings from Bei-chuan and Hong-ya bases have been introduced to other provinces in China. To date, more than 150 yew forest farms have been established in various provinces of China, some of which can provide suitable active pharmaceutical ingredients [28].

Typically, for the extraction of paclitaxel or its precursor 10-deacetylbaccatin III (10-DAB), the branches and/or needles are harvested to keep the plant alive. This method can prevent the destruction of wild resources and achieve the sustainable use of resources. Paclitaxel is easily soluble in organic solvents, so ethanol, methanol, chloroform, ethyl acetate-acetone and ionic liquids can be used to extract paclitaxel from the plant. In recent years, microwave-assisted solvent extraction (MASE), ultrasound-assisted extraction (USAE), supercritical CO_2_ extraction and pressurized solvent extraction (PLE) have been widely used for the extraction of paclitaxel [46,47,48,49,50,51]. These methods can reduce the amount of solvent and operating time required and increase the purity and yield of paclitaxel compared to that achieved with conventional extraction methods. For example, Min et al. utilized the synergistic effect of ultrasound and negative pressure cavitation extraction (NPCE) to achieve more than 99% extraction of paclitaxel at an ultrasound power of 380 W and a vacuum of 260 mm Hg, with an extraction time of only 3 min [52].

Currently, artificial propagation of *Taxus* seedlings is considered one of the most efficient methods for obtaining paclitaxel and its chemical semi-synthetic precursors. Meanwhile, the extraction technology for paclitaxel is improving. However, these do not fully address the inadequate supply of paclitaxel.

### 3.2. Total Synthesis

Total synthesis is ideal for addressing the clinical supply of paclitaxel. However, the complex structure is a major obstacle to total synthesis of paclitaxel. Paclitaxel possesses a highly oxygenated [6-8-6-4] core with 11 stereocenters. Moreover, the unique bicyclo [5.3.1] undecane ring system, densely aligned oxygen functionalities and four flanking acyl groups all contribute to heightening the challenge of its chemical construction. In 1994, Nicolaou et al. and Holton et al. reported the first total synthesis of paclitaxel, and subsequently, various total synthesis methods were reported [53,54,55]. So far, eleven total syntheses and three formal syntheses, as well as over 60 synthetic model studies of paclitaxel, have been completed by more than 60 research groups worldwide [56,57]. The eight-membered ring synthetic strategy for each total synthesis is summarized in Figure 4, and the starting materials, key steps and total steps for each total synthesis are summarized in Table 3. In the synthesis of paclitaxel (Figure 4), bonding to close the required eight-membered ring usually takes place at the top of the planar structure of the molecule. In particular, the top C9–C10 bond disconnections (Nicolaou, Kuwajima and Takaihashi) and the C10–C11 bond disconnections (Danishefsky, Kishi, Chida and Nakada) are the most used (7 out of 14 syntheses). It is noteworthy that both Holton and Wender used Grob-type fragmentation to construct the A and B rings; Mukaiyama’s total synthetic approach was based on an intramolecular aldol cyclization employing SmI_2_ to synthesize the eight-membered ring at the C3–C8 site; Baran utilized the type II IMDA reaction to form the A and B rings through the formation of C1–C15 and C13–C14 bonds; and Li and Inoue utilized SmI_2_–pinacol coupling to form the eight-membered ring through the formation of C1–C2 bonds. These successful methods for the total synthesis of paclitaxel are landmarks in the field of organic chemistry. In general, the pathway of the full chemical synthesis method for paclitaxel is too long and there are too many synthetic steps. Not only are expensive chemical reagents required, but the reaction conditions are also difficult to control and the yield is low (e.g., the overall yield of Li’s 21-step synthetic route was 0.118%), which is not suitable for industrial production. Further efforts are needed to reduce the synthesis steps and improve the yield of paclitaxel total synthesis.

### 3.3. Semi-Synthesis

From the 1960s to the 1980s, paclitaxel could only be isolated from the bark of the yew tree in very low yields. In 1988, Dr. Denis first obtained 10-deacetylbaccation III (10-DAB) from yew needles and used it for the semi-synthesis of paclitaxel with a 53% yield [71]. Subsequently, Prof. Holton and Prof. Potier patented the semi-synthesis of paclitaxel from baccatin III. The US Bristol Myers Squibb (BMS) company received approval from the FDA to produce paclitaxel using Prof. Holton’s patent for the semi-synthesis of paclitaxel from baccatin III and decided to discontinue the extraction of paclitaxel from the bark of the yew tree at the end of 1994.

The chemical semi-synthesis of paclitaxel is the main source of paclitaxel in the current market, accounting for approximately 80% of the market share. Because of the relatively high content of 10-DAB and baccatin III in the needles and twigs of *Taxus*, studies on the semi-synthesis of paclitaxel mainly focus on these two substances. More than twenty routes for the semi-synthesis of paclitaxel have been reported [72], and three different types of side chains, including linear phenylisoserine, β-lactam tetracyclic and oxazolidine pentacyclic, are primarily utilized to react with 7-triethylsilyl baccatin III (7-TES-baccatin III), which is then deprotected to produce paclitaxel. It should be noted that the side chain C2′-OH is prone to epimerization during chemical synthesis, which not only greatly affects the yield of paclitaxel but also directly pushes up the production cost of paclitaxel. The semi-synthesis method of paclitaxel by BMS [73], common types of paclitaxel semi-synthetic side chains and advantages and disadvantages of different side-chain syntheses are shown in Figure 5. In addition, Borah et al. comprehensively summarized the methods for the synthesis of C-13 chiral side chains in 2007, including asymmetric epoxidation routes, enol–imine condensation, the Diels–Alder reaction, β-lactams and the use of asymmetric catalysts (Figure 6, Figure 7 and Figure 8) [74,75].

In addition to 10-DAB and baccatin III, which are the two commonly used semi-synthetic raw materials, other natural precursors such as 10-deacetyl-7-xylosyltaxanes and 10-deacetyl paclitaxel-7-xyloside have also been reported to be converted to paclitaxel. In the cultivated *Taxus*, 10-deacetyl-7-xylosyltaxanes (a mixture of 10-deacetyl-7-xylosyltaxols A, B and C), which is 10–30 times more abundant than paclitaxel, is usually transformed to 10-DAB or discarded as waste [76]. In 2020, Xue et al. found that these compounds can be converted to paclitaxel through a three-step reaction of redox, acetylation and deacetylation with a purity of 99.52% with 67.6% total yield (Figure 9a). This synthetic process circumvented the use of 10-DAB precursors and expensive chiral side chains, resulting in lower costs, fewer reaction steps and significantly higher yields [77]. 10-Deacetyl paclitaxel-7-xyloside (XDT) was isolated from the bark of *Taxus brevifolia* and has a structure similar to that of paclitaxel. After hydrolysis of the xylose moiety at the C-7 site, paclitaxel can be obtained by a three-step reaction of TES protection, acetylation and deprotection of TES (Figure 9b) [78]. Compared to 10-DAB or baccatin III precursors for semi-synthetic methods, this precursor contains a C-13 side chain and has a simple synthesis procedure.

### 3.4. Tissue and Cell Culture

Plant cells are totipotent, and the induction and regulation of paclitaxel synthesis in yew cells is a current research hotspot for paclitaxel drug development. Plant cell culture can completely alleviate the dependence on the *Taxus* plant and mitigate the effects of survival conditions such as temperature on paclitaxel yield. Moreover, it not only avoids the complex transgenic manipulations but also avoids the introduction of exogenous genes that produce cytotoxicity. Currently, two companies, Phyton Biotech (U.S.) and Samyang Genex (Korea), supply paclitaxel extracted from cultured plant cells, accounting for about 10% of the paclitaxel market share.

Christen et al. first discovered that the cell cultures of *Taxus brevifolia* could produce paclitaxel in 1989, a finding that was patented two years later [79]. The production of paclitaxel can reach 1–3 mg/L within 2 to 4 weeks. Currently, more than ten *Taxus* species or variants have been found to produce paclitaxel and paclitaxel-like compounds. However, paclitaxel was found in low and often unstable yields in cultured *Taxus* cells, which hampered large-scale production. Therefore, various factors affecting cell suspension culture such as *Taxus* species, culture conditions, phytohormones and inducers have been widely studied [1,80,81,82]. For example, significantly increased amounts of paclitaxel (28 to 110 mg/L) were observed in cell cultures of *Taxus* species by adding methyl jasmonate [83]. Wang et al. screened three stable, high-yielding cell lines from *Taxus cuspidate*, and they are promising candidate sources for the large-scale production of paclitaxel [84].

In addition to different *Taxus* species, other natural sources for paclitaxel production have also been explored, such as *Corylus avellana*, the hazelnut tree. Although the concentration of paclitaxel in *C. avellana* is 10 times lower than that in yews [85], *C. avellana* cells grow faster, and the paclitaxel content of cell suspension cultures of *C. avellana* as well as the excretion of paclitaxel in the culture medium could be increased by treatments with inducers such as methyl jasmonate. Gallego et al. found that treatment of *C. avellana* cells with methyl jasmonate (100 μM) and coronarin (1 μM) exciton increased the paclitaxel content in cell suspension cultures 3-fold and 27-fold, respectively [86]. The cell wall of *C. palmarum* was the most effective fungal inducer of paclitaxel synthesis in *C. avellana* cell medium. The combination of the cell wall of *C. palmarum* and methyl-β-cyclodextrin (50 mM) as an inducer increased the total production of paclitaxel in *C. avellana* cell medium 5.8-fold (402.4 μg/L), of which 78.6% (316.5 μg/L) was secreted into the medium [87].

### 3.5. Paclitaxel-Producing Endophytic Fungi

Endophytic fungi are present in plants and co-evolve with their host plants. They produce biologically active secondary metabolites that are identical or similar to those made by the host plants. Thus, endophytic fungi of plants can be a new platform for the commercial production of bioactive metabolites. In 1993, the endophytic fungus *Taxomyces andreanae* from *T. brevifolia* was discovered by Stierle et al. to produce paclitaxel in vitro [88]. Since then, more and more researchers have been engaged in isolating and characterizing paclitaxel-producing endophytic fungi [89,90,91]. To date, more than 20 genera of endophytic fungi have been identified in *Taxus* species and non-*Taxus* species such as sycamore and ginkgo (Table 4) [92,93,94,95,96,97,98,99,100,101,102,103,104,105,106,107,108,109,110,111,112,113,114,115,116,117,118,119,120,121,122,123,124,125,126,127,128,129,130,131,132,133,134,135,136,137,138,139,140,141]. In addition to plants, paclitaxel-producing endophytic fungi can also be isolated from animals. In 2015, Gu et al. first isolated *Pestalotiopsis hannanensis* from the scalp of *Ailuropoda melanoleuca*, a giant panda with skin disease, which produced paclitaxel at a yield of 1466.87 μg/L [142]. The production of paclitaxel from endophytic fungi by microbial fermentation is a sustainable way of obtaining paclitaxel. This method is characterized by simple medium formulation, controlled fermentation conditions and mature technology for large-scale production. However, the content of paclitaxel obtained in this way is generally low. Therefore, there is still a long way to go for the industrial production of paclitaxel using endophytic fungi in the future.

In recent years, optimization of the fermentation culture is one of the important ways to increase paclitaxel production by endophytic fungi through complementation with a variety of substances including carbon sources, nitrogen sources, precursors, elicitors and metabolic bypass inhibitors.

Garyali et al. isolated the endophytic fungus *Fusarium redolens* from Himalayan yew plants and demonstrated its ability to produce paclitaxel. The results showed that sucrose and NH_4_NO_3_ were the best carbon and nitrogen sources for paclitaxel production. The yield of paclitaxel synthesized by *Fusarium redolens* increased from 66 to 198 μg/L with the addition of NH_4_NO_3_ (6.25 g/L), MgSO_4_·7H_2_O (0.63 g/L) and NaOAc (1.25 g/L) to the medium, which was three times higher than the yield in unoptimized medium [112,143]. Furthermore, addition of early precursors (isopentenyl pyrophosphate (IPP) and geranylgeranylpyrophosphate (GGPP)) of the terpene pathway to cell cultures of the endophytic fungus *Paraconiothyrium* SSM001 plants stimulated terpene production, with a 3-fold and 5-fold increase in the production of paclitaxel compared to controls [144]. Qiao et al. isolated a strain of *A. aculeatinus* from *Taxus* bark and confirmed that the endophytic fungus *A. aculeatinus* Tax-6 was able to produce paclitaxel in potato dextrose agar liquid medium. Since sodium acetate is an important precursor of paclitaxel, Cu^2+^ can enhance the activity of oxidase, thereby catalyzing the formation of paclitaxel, and salicylic acid can act as an induction signal. The introduction of Cu^2+^ (0.1 mg/L), salicylic acid (10 mg/L) and NaOAc (8 g/L) to the medium increased the yield of paclitaxel from 334.92 μg/L to 1337.56 μg/L g/L [145].

In addition, co-culture of *B. subtilis* and *A. flavipes* can regulate paclitaxel biosynthesis in *A. flavipes* by modulating chromatin remodeling, resulting in an approximately 1.6-fold increase in paclitaxel production [146]. When fluconazole (1.0 μg/mL) was co-cultured with *A. flavipes*, paclitaxel production was increased 5-fold [110,147]. When salicylic acid and *P. microspora* were co-cultured, the yield of paclitaxel was 625.47 μg/L, which was 45 times higher than that of the control group. This is due to the fact that salicylic acid enhances the lipid peroxidation reaction in *P. microspora* mycelia, and the production of peroxides stimulates oxidative stress, which induces the activation of 3-hydroxy-3-methyl glutaryl coenzyme A reductase (HMGR) proteins by regulatory proteins and eventually triggers the expression of GGPSG to stimulate the isoprenoid biosynthetic pathway, leading to improved biosynthesis in *P. microspora* [148].

The pH of the culture medium also has an effect on the yield of paclitaxel. El-Sayed et al. isolated *Penicillium chrysogenum* strains from the inter-root region of *Glycine max*, a legume that can produce paclitaxel, and investigated the effect of initial pH on the growth and paclitaxel production of *P. chrysogenum*. The results showed that paclitaxel production reached a maximum of 200–220 μg/L at a pH of 7–8 at a temperature of 30 °C and an agitation rate of 120 rpm [149]. Yang et al. isolated and characterized *Alternaria alternata* MF5 to produce paclitaxel. The results showed that the production of paclitaxel started at 12 h (1.193 mg/L, pH = 6.21) and reached a maximum value at 60 h (pH = 4.96), and the production gradually decreased after 60 h. A pH of 4.8–5.2 is the optimal pH for rpm production [150]. However, Abdel-Fatah et al. optimized the yield of paclitaxel production from *Aspergillus flavus* by CCD design and found that the maximum yield of paclitaxel (302.72 μg/L) was achieved at pH 6.0 when other conditions were the same [151].

Temperature also effects fungal growth and paclitaxel synthesis. For example, the maximum radial growth of *Fusarium solani* was achieved at 30 °C [152]. The yield of paclitaxel is also influenced by light. Under natural conditions, plant tissues provide protection to endophytic fungi, so the fungi do not need to produce pigments and instead use their metabolic resources to produce paclitaxel against fungal pathogens of the host plant. However, once the fungus is released and exposed to light, the endophytic fungus shifts its metabolic resources from the synthetic paclitaxel pathway to the production of defensive pigments. Thus, paclitaxel production by endophytic fungi increases when plant host conditions are simulated [153,154].

## 4. Synthetic Biology Studies of Paclitaxel

In recent years, with the successful application of synthetic biology in the synthesis of natural products, synthetic biology research on paclitaxel has also attracted much attention, and the work in this area mainly focuses on the analysis of the biosynthetic pathways of paclitaxel and the construction and optimization of the precursor cell factory of paclitaxel.

### 4.1. Biosynthetic Pathways

Paclitaxel has a complex molecular structure, and its biosynthetic pathway is equally complicated. Until today, the biosynthetic pathway of paclitaxel is not yet fully understood, as several steps remain undetermined and several enzymes remain unknown. The pathway is postulated to involve 19 steps and is divided into three parts: (1) synthesis of paclitaxel precursor 10-DAB or baccatin III from GGPP, a precursor of diterpene compounds; (2) synthesis of the phenyl-isoserine side chain; (3) acylation linkage of the side chain and the C-13 position of baccatin III to form paclitaxel by hydroxylation and benzoylation (Figure 10) [155].

There is a large number and variety of enzymes involved in the first stage, which is key to the formation of paclitaxel. Taxadiene synthase (TS) catalyzes the first and committed step to cyclize GGPPS to taxa-4(5),11(12)-diene (taxadiene). After the formation of the backbone, further hydroxylation at the C1, C2, C5, C7, C9, C10 and C13 sites is followed by acylation, carbonylation, epoxidation and benzoylation, ultimately resulting in the formation of baccatin III, the precursor compound of paclitaxel. Among the enzymes, five CYP450s [taxane 5α-hydroxylase (T5αH), taxane 2α-hydroxylase (T2αH), taxane 7β-hydroxylase (T7βH), taxane 10β-hydroxylase (T10βH) and taxane 13α-hydroxylase (T13αH)] responsible for catalyzing the hydroxylation of C-2, C-5, C-7, C-10 and C-13 sites have been cloned and identified. Enzymes that remain unidentified are the CYP450s taxane 1β-hydroxylase (T1βH), taxane 9α-hydroxylase (T9αH), taxane 9α-oxidase (T9αO) and C4,5 epoxidase.

The synthesis of the C13 side chain is a key factor in ensuring the anticancer activity of paclitaxel [13] and is accomplished through a two-step reaction: α-phenylalanine isomerizes to form β-phenylalanine catalyzed by phenylalanine aminomutase (PAM), and then β-phenylalanine-CoA forms with acetyl coenzyme A in the presence of phenylalanine-CoA ligase (PCL) [156].

Subsequently, β-phenylalanyl-CoA is catalyzed by C-13 phenylpropanoyl-CoA transferase (BAPT) to form β-phenylalanyl baccatin III. The latter is further hydroxylated to form 3′-*N*-debenzoyltaxol by the action of taxane 2′α-hydroxylase (T2′αH), which was recently isolated from mining the *T. baccata* transcriptome [157]. Finally, paclitaxel was obtained by the benzoylation of the nitrogen atom at the C3′ site of the side chain under the catalysis of debenzoyl taxol *N*-benzoyl transferase (DBTNBT).

### 4.2. Ab Initio Biosynthesis of Paclitaxel by Heterologous Systems

The production of paclitaxel in more amenable, fast-growing, heterologous hosts is a truly sustainable green pathway, as there is no dependence on *Taxus* species at all. In recent years, significant advances were conducted to develop heterologous systems for paclitaxel biosynthesis, leading to the accumulation of paclitaxel intermediates. Recent achievements in different hosts are summarized in Table 5.

In 2001, Huang et al. realized the first heterologous synthesis of taxadiene by co-expression of 1-deoxy-D-xylulose 5-phosphate synthase (DXS), isopentenyl pyrophosphate isomerase (IDI), geranylgeranyl diphosphate synthase (GGPPS) and TS in a single strain of *E. coli* with an unoptimized yield of 1.3 mg per liter of cell culture [160]. In 2010, Ajikumar et al. reported a multivariate modular approach to metabolic pathway engineering by which the biosynthetic pathway of paclitaxel was divided into two parts: a natural mevalonate (MEP) pathway leading to IPP and dimethylallyl pyrophosphate (DMAPP), and a downstream terpene synthesis route. By optimally balancing the two blocks, the yield of paclitaxel in engineered *E. coli* was eventually successfully increased 15,000-fold to ~1 g/L. And with the subsequent introduction of T5αH and the *Taxus* CYP450 reductase, taxadiene-5α-ol was heterologously synthesized with a yield of 58 mg/L [158]. *E. coli* is an excellent host, but P450s are hard to express in *E. coli* due to the lack of an endomembrane system. In 2016, Biggs et al. achieved efficient expression of T5αH in *E. coli* by optimizing P450 expression, *N*-terminal modification and reductase chaperone interaction, with an oxygenated taxane yield of 570 ± 45 mg/L [159].

*S. cerevisiae* is also a common chassis for the heterologous synthesis of natural products. *S. cerevisiae* produces a functional type II P450 monooxygenase with an intact intracellular membrane system that ensures the co-expression of hydroxylase genes associated with paclitaxel biosynthesis [179]. Therefore, *S. cerevisiae* is more suitable and feasible for paclitaxel intermediate expression. In 2008, Engels et al. efficiently synthesized taxadiene in *S. cerevisiae* by establishing an adequate supply of GGPP and significantly increased TS expression through codon optimization. Finally, taxadiene (8.7 mg/L) and geranyl geraniol (33.1 mg/L) were obtained, which was the first demonstration of such enhanced taxadiene levels in *S. cerevisiae*, indicating that taxadiene levels could be further increased [164]. Ding et al. constructed a pathway for paclitaxel biosynthesis by overexpressing ERG20 and tHMGR genes in *S. cerevisiae* and introducing TS genes, and paclitaxel yield reached 72.8 mg/L [165]. Zhou et al. reported a co-culture method for the production of oxygenated taxanes using *E. coli* and *S. cerevisiae* [161]. *E. coli* was used for taxadiene production, whereas *S. cerevisiae* was employed for acetylation and CYP450-oxygenation chemistry. This study combined the strengths of *E. coli* and *S. cerevisiae* and demonstrated the feasibility of microbial consortia to rebuild the metabolite pathway.

Plant systems are safer and more economical than microbial systems. In 2019, Li et al. utilized chloroplastic metabolic engineering to express TS, T5αH and cytochrome P450 reductase in *Nicotiana benthamiana* and successfully obtained taxadiene and taxadiene-5α-ol with yields of 56.6 μg/g and 1.3 μg/g, respectively [174]. This study shows that tobacco is a potential heterologous plant platform for the production of paclitaxel and lays the foundation for further synthesis of oxygenated taxanes. The use of plant systems for the synthesis of plant-derived natural products has theoretical advantages, such as its ability to produce secondary metabolites from sunlight and natural carbon dioxide. However, the superiority of plant systems over microbial systems in terms of culture conditions, difficulty of genetic manipulation and mass cultivation is not prominent. Therefore, the realization of heterologous synthesis of paclitaxel or its key precursors in plant chassis remains a long way to go.

### 4.3. Semi-Synthesis by Microbial Systems

Paclitaxel analogues are structurally similar to paclitaxel and can be converted to paclitaxel in just a few steps. Among them, XDT is the most abundant paclitaxel analogue in the bark of the yew tree. The amount of XDT (0.4% of dry weight) is much higher than that of paclitaxel. While XDT is not an intermediate in the paclitaxel biosynthetic pathway, it can be transformed to paclitaxel via deglycosylation and acetylation. However, it is often discarded in the process of extracting paclitaxel, resulting in the waste of resources and potential environmental pollution. In 2017, Li et al. improved the catalytic efficiency of 10-deacetylbaccatin III-10-Oacetyltransferase (DBAT) of *Taxus* by mutagenesis and then combined DBAT with a β-xylosidase to obtain an in vitro one-pot conversion of XDT to paclitaxel yielding 0.64 mg/mL paclitaxel in 50 mL at 15 h (Figure 11) [180]. This approach shows a promising, eco-friendly alternative for paclitaxel production from an abundant analogue. However, the precursors currently used are still of plant origin, and the paclitaxel supply issue is essentially not yet fully solved.

In summary, paclitaxel has been extensively studied in the past decades both from the biosynthetic and chemosynthetic standpoints. Although these methods continue to evolve, there are still inevitable problems that limit their capabilities and drive up the price of paclitaxel (Table 6). Therefore, the development of efficient methods for the production of paclitaxel is highly desirable.

## 5. Conclusions

As the tumor incidence rate around the world is increasing, malignant tumors such as lung, breast and ovarian cancers have also become more prevalent, and affected patients are the main users of paclitaxel. On the other hand, paclitaxel has been found to have other medical uses beyond anticancer drugs. For instance, paclitaxel gel, a topical formulation of paclitaxel for the treatment of rheumatoid arthritis, has been developed and marketed in the United States. In addition, paclitaxel has been used as a coating agent for vascular stents in medicinal devices. Newly developed uses of paclitaxel have further boosted the demand for paclitaxel crude drugs on the international market.

Given this information, there has been a worldwide effort to address the availability of paclitaxel over the past several decades. Although chemical semi-synthesis and direct extraction of paclitaxel from the nursery cultivation of *Taxus* species are the main sources for the clinical supply of paclitaxel, they are still dependent on plant material, and the problem of paclitaxel supply is not inherently fully solved. Total synthesis research showed that it was feasible to prepare paclitaxel in the laboratory and laid the foundation for future approaches to paclitaxel. However, it remained within the realm of academic research. The isolation of paclitaxel from endophytic fungi by microbial fermentation is considered a sustainable method for obtaining paclitaxel, but no breakthroughs have been made. Tissue and cell cultures represent an alternative and environmentally sustainable source of paclitaxel. To increase paclitaxel yield, efforts have been made to optimize culture conditions, screen highly productive cell lines and induce secondary metabolite pathways. Future perspectives should be concentrated on the simultaneous use of empirical and rational approaches.

Synthetic biology methods have been widely used for biosynthetic research on paclitaxel. Currently paclitaxel precursors such as taxadiene have been synthesized heterologously in microbial and plant systems, but further studies are needed to understand the missing pathway enzymes and regulatory mechanisms. With the sequencing of the *Taxus* genome, as well as progress in enzyme engineering, the biotechnological production of paclitaxel will no longer be a dream in the near future.

## Figures and Tables

**Figure 1 molecules-28-07517-f001:**
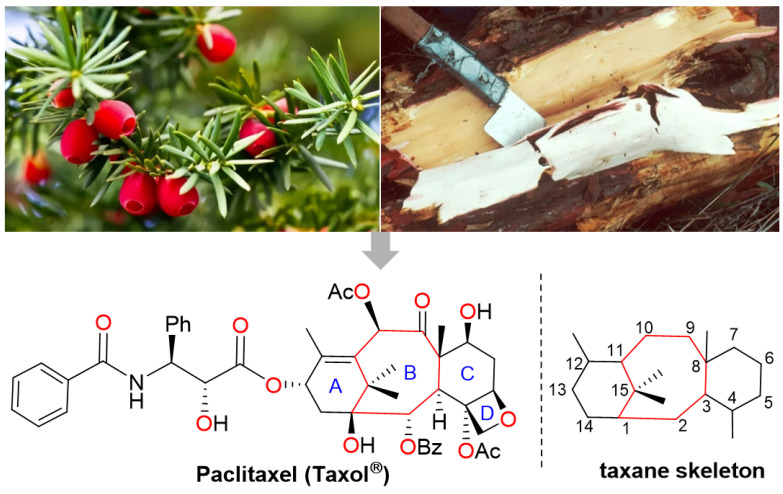
The structure of paclitaxel.

**Figure 2 molecules-28-07517-f002:**
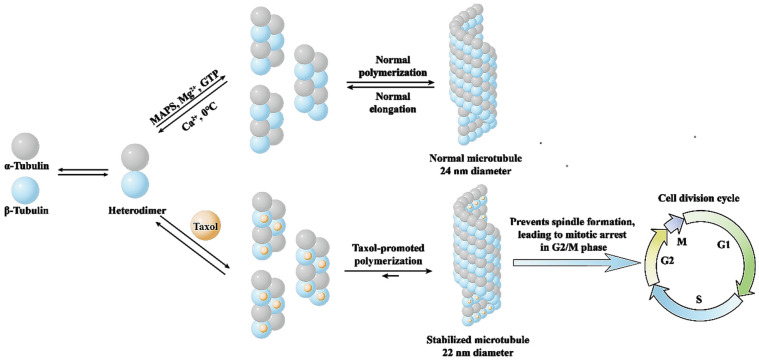
Mechanism of action of paclitaxel on microtubulin.

**Figure 3 molecules-28-07517-f003:**
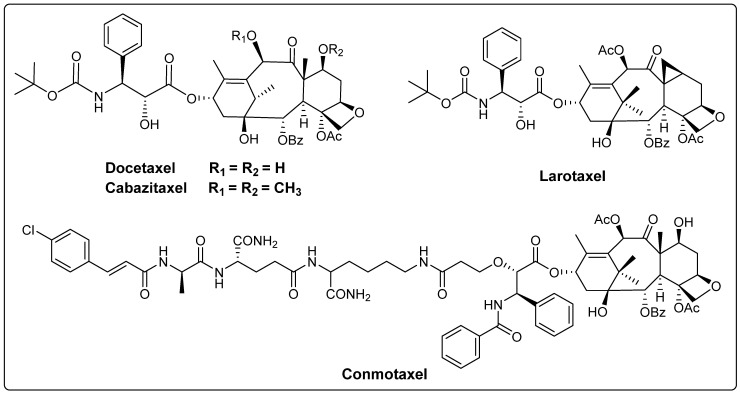
The structures of paclitaxel-like compounds.

**Figure 4 molecules-28-07517-f004:**
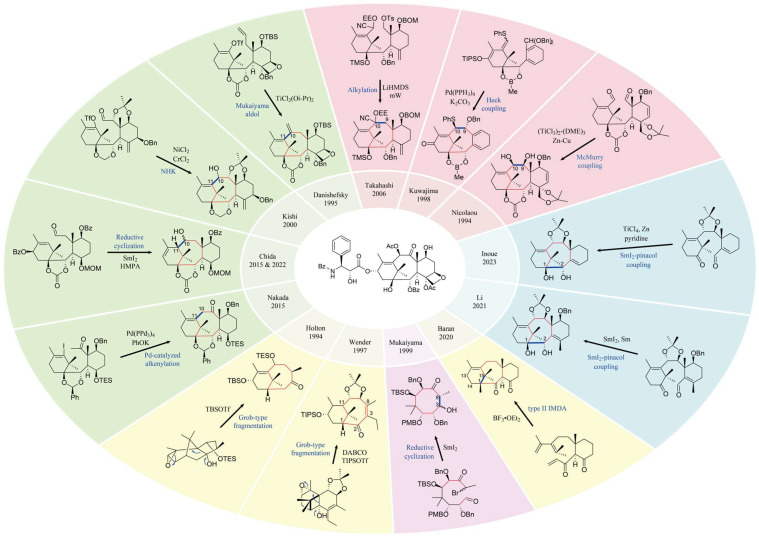
Summary of the eight-membered ring synthetic strategy of paclitaxel [53,54,55,56,58,59,60,61,62,63,64,65,66,67,68,69,70].

**Figure 5 molecules-28-07517-f005:**
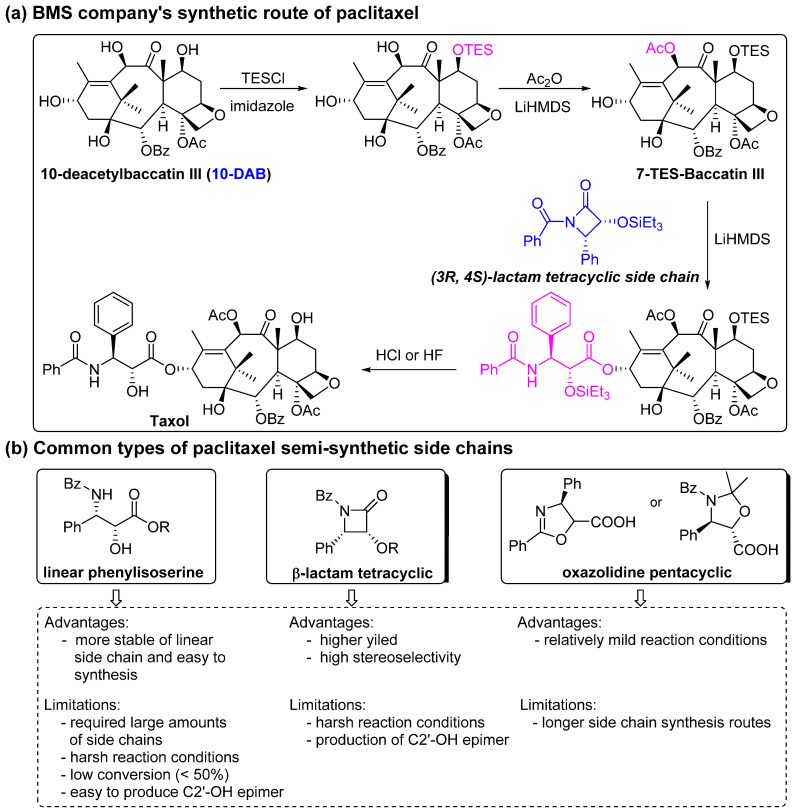
BMS company’s synthetic route for paclitaxel and common types of paclitaxel semi-synthetic side chains [57,73].

**Figure 6 molecules-28-07517-f006:**
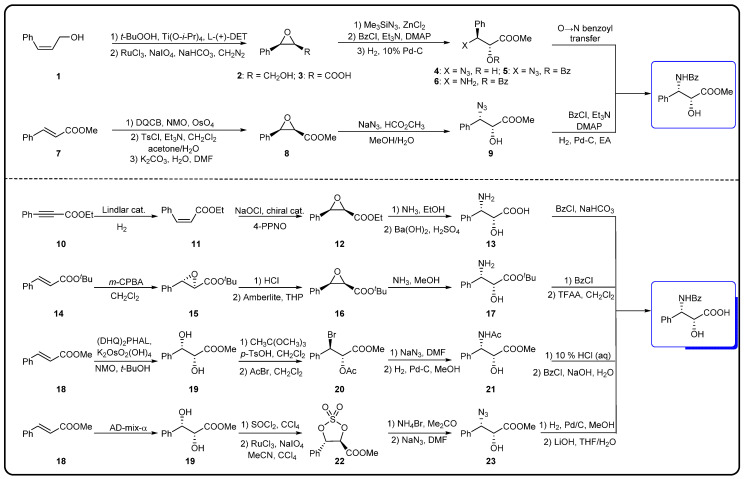
Synthesis of linear phenylisoserine side chains [74].

**Figure 7 molecules-28-07517-f007:**
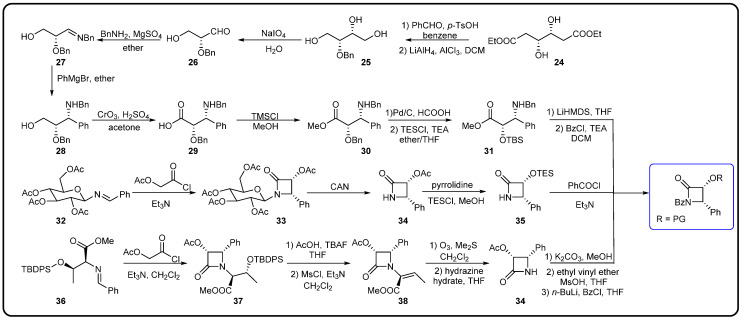
Synthesis of β-lactam tetracyclic side chains [74].

**Figure 8 molecules-28-07517-f008:**
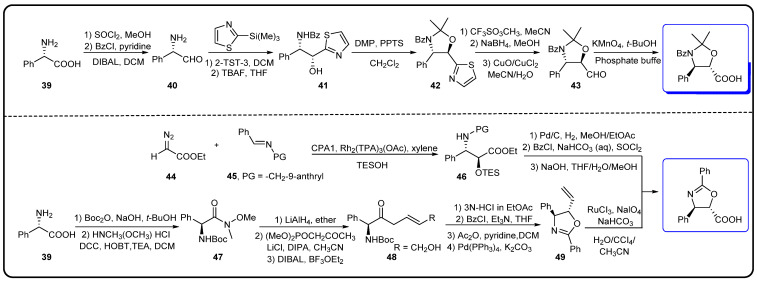
Synthesis of oxazolidine pentacyclic side chains [74,75].

**Figure 9 molecules-28-07517-f009:**
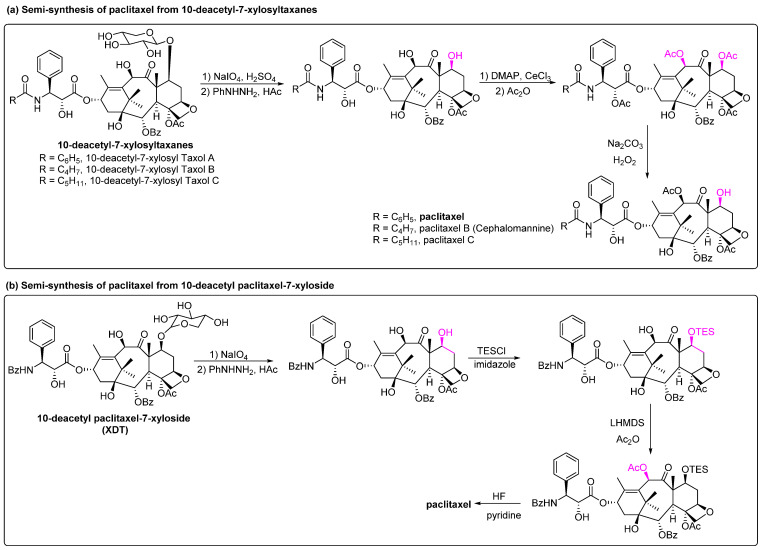
Semi-synthesis of paclitaxel from 10-deacetyl-7-xylosyltaxanes and 10-deacetyl paclitaxel-7-xyloside [77,78].

**Figure 10 molecules-28-07517-f010:**
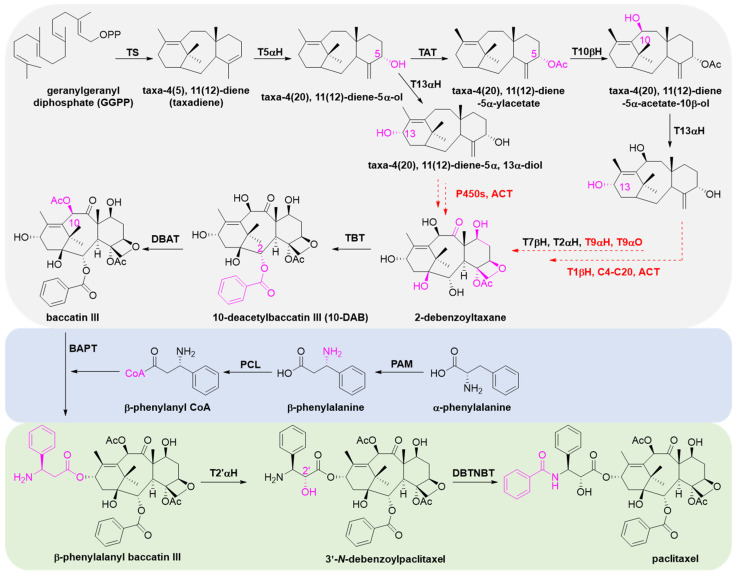
The biosynthetic pathway of paclitaxel. Enzymes in red are not yet characterized, and steps in red dotted arrows are not yet fully elucidated. Enzyme abbreviations: TS, taxadiene synthase; T5αH, taxane-5α-hydroxylase; TAT, taxane-5α-ol-*O*-acetyltransferase; T10βH, taxane-10β-hydroxylase; T13αH, taxane-13α-hydroxylase; T1βH, taxane 1β-hydroxylase; T9αH, taxane 9α-hydroxylase; T9αO, taxane 9α-dioxygenase; T2αH, taxane 2α-hydroxylase; T7βH, taxane 7β-hydroxylase; C4-C20, C4-C20 epoxidase; TBT, taxane-2α-O-benzoyl transferase; DBAT, 10-deacetylbaccatin III-10-*O*-acetyltransferase; PAM, phenylalanine aminomutase; PCL, phenylalanine-CoA ligase; BAPT, C-13 phenylpropanoyl-CoA transferase; T2′αH, taxane 2′α-hydroxylase; DBTNBT, debenzoyl taxol *N*-benzoyl transferase; ACT, acyl-CoA transferase.

**Figure 11 molecules-28-07517-f011:**
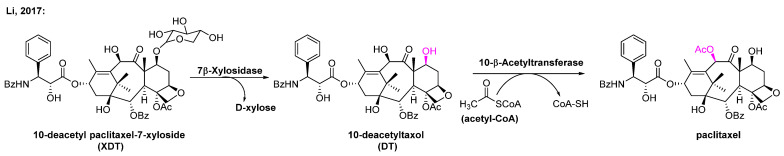
One-pot reaction system for the bioconversion of XDT to paclitaxel. The system contained a specific 7-β-xlyosidase, the improved 10-β-acetyltransferase, the substrate XDT and the acetyl group donor acetyl-CoA [180].

**Table 1 molecules-28-07517-t001:** Clinical combinations of paclitaxel.

Condition or Disease	Partner Drugs	Condition or Disease	Partner Drugs
Lung cancer	Cisplatinum	Lymphoma	Cisplatinum
Gastric cancer	Tegafur		Adriamycin
	Capecitabine		Capecitabine
Head and neck tumors	Cisplatinum		Gemcitabine
Esophageal cancer	Cisplatinum		Adriamycin + cyclophosphamide
	Capecitabine	Pancreatic	Gemcitabine

**Table 2 molecules-28-07517-t002:** Ongoing clinical trials of paclitaxel in combination with other drugs.

Partner Drugs	Condition or Disease	Phase	Clinical Trial Identifier
Gemcitabine	Refractory solid tumors	I	NCT03507491
Raltitrexed	Advanced pancreatic cancer	II	NCT04581876
Apatinib and camrelizumab	Advanced gastric cancer	I/II	NCT04286711
AZD2014	Advanced cancer	I	NCT02193633
Fostamatinib	Ovarian cancer	I	NCT03246074
Tilelizumab	High-risk non-muscle-invasive urothelial bladder carcinoma that is not completely resectable	II	NCT04730232
LDE225	Recurrent ovarian cancer	I	NCT02195973
Camrelizumab	Non-small cell lung cancer	II	NCT04167774
Gemcitabine and ficlatuzumab	Pancreatic cancer	I	NCT03316599
Durvalumab	Squamous cell carcinoma of the head and neck	II	NCT03723967
fruquintinib	Gastric cancer	III	NCT03223376
Chiauranib	Ovarian cancer	III	NCT04921527
Lovastatin	Ovarian cancer	II	NCT00585052
Capibasertib	Locally advanced (inoperable) or metastatic triple-negative breast cancer	III	NCT03997123
Pembrolizumab and carboplatin	Recurrent/metastatic head and neck squamous cell carcinoma	IV	NCT04489888

This data was obtained from the International Center for Clinical Trials Research (https://clinicaltrials.gov/, accessed on 22 May 2023).

**Table 3 molecules-28-07517-t003:** Summary of the total synthetic route of paclitaxel.

Research Groups	Year	Synthetic Strategy	Starting Materials	Total Steps	Refs.
Nicolaou et al.	1994	(1) Coupling of the A and C rings by Shapiro reaction at the C1–C2 positions; (2) formation of the B ring by McMurry coupling at the C9–C10 positions; (3) and finally, formation of the side chain, selective oxidation of the C13 position and formation of the D ring.	Ethyl 4-hydroxy-2-methylbut-2-enoate and 3-hydroxy-2-pyrone	51	[53]
Holton et al.	1994	(1) Formation of AB ring by epoxy alcohol cleavage; (2) formation of the C ring by Dieckmann condensation; (3) formation of the D ring based on intramolecular *S*_N_2 cyclization; (4) introduction of the C9 oxygen functional group and side chain.	Camphor	41	[54,55]
Danishefsky et al.	1996	(1) Coupling of A and CD rings at the C1–C2 sites via 1,2-addition reactions; (2) generation of B rings via Heck coupling at the C9–C10 sites; (3) selective oxidation of C9 and C13 and formation of side chains.	2-Methyl-1,3-cyclohexandione	47	[58]
Wender et al.	1997	(1) Formation of the AB ring by Grob-type fragmentation; (2) formation of the C ring by aldol cyclization reaction; (3) formation of the D ring based on intramolecular S_N_2 cyclization and introduction of the side chain.	Verbenone	37	[59,60]
Kuwajima et al.	1998	(1) Coupling of the A ring and C ring at the C1–C2 site by 1,2-addition reaction; (2) generation of the B ring at the C9–C10 site by Vinylogous Mukaiyama aldol reaction; (3) formation of the D ring and C13 side chain by introducing C19-methyl and the C3 standing center.	2-(Prop-2-yn-1-yloxy)tetrahydro-2H-pyran	47	[61]
Mukaiyama et al.	1999	(1) Formation of octacycles at the C3–C8 sites by intramolecular aldol cyclization of SmI_2_; (2) formation of C rings at the C7–C8 sites based on Michael addition and intramolecular hydroxyl aldol cyclization; (3) formation of A rings at the C11–C12 sites based on pinacol coupling cyclization; (4) final selective oxidation of C13 as well as the formation of D rings and side chains.	Methyl 3-hydroxy-2,2-dimethylpropanoate	38	[62]
Kishi et al.	2000	(1) Introduction of a C8 all-carbon quaternary center by [2,3] rearrangement; (2) Coupling of A and C rings at the C1–C2 sites by 1,2-addition reaction; (3) generation of the B ring by NHK coupling at C9–C10 sites; (4) formation of the D ring and side chain, oxidation of C13.	3-Methylcyclohex-2-en-1-ol	45	[56]
Takahashi et al.	2006	(1) Coupling of the A ring and CD ring at the C1–C2 sites by 1,2-addition reaction; (2) generation of the B ring by microwave-assisted alkylation at the C9–C10 sites; (3) selective oxidation of C9 and formation of the D ring.	Geraniol	47	[63]
Nakada et al.	2015	(1) Coupling of the A and C rings at the C1–C2 sites based on 1,2-addition reactions; (2) generation of the B ring by palladium-catalyzed alkenylation at the C9–C10 sites; (3) formation of the D ring by S_N_2 cyclization.	Acetal aldehyde and vinyl iodide	37	[64]
Chida et al.	2015	(1) Linking the A and C rings by 1,2-addition at the C1–C2 site; (2) forming the B ring by palladium-catalyzed alkenylation at the C10–C11 bond; (3) constructing the D ring of oxetane by S_N_2 cyclization.	*Tri*-*O*-acethl-D-glucal and 1,3-cyclohexanedione	42	[65,66]
Baran et al.	2020	(1) Type II intramolecular Diels–Alder reaction to form the ABC framework; (2) stereoselective oxidation to C13, C5, C10 and C9 sites; (3) dioxane-mediated C–H oxidation to produce bridging tertiary alcohols at the C1 site; (4) formation of the D ring and side chain.	2,3-Dimethylbut-2-ene; 3-ethoxy-2-cyclohexen-1-one; CHBr_3_; acrylaldehyde	24	[67]
Li et al.	2021	(1) Asymmetric synthesis to form the AC ring; (2) SmI_2_-mediated pinacol coupling reaction to form the B ring; (3) generation of C3 stereocenters by the Hutchins–Kabalka method; (4) formation of the D ring as well as the introduction of the C13 side chain.	(2*R*, 3*S*)-2-Allyl-3-hydroxy-2-methylcyclohexan-1-one	21	[68]
Chida et al.	2022	(1) Linking the A and C rings by 1,2-addition at the C1–C2 site; (2) forming the B ring by palladium-catalyzed allylation at the C10–C11 bond; (3) forming the C13 and C5 hydroxyl groups by Rubottom oxidation; and (4) forming the D ring by a novel sliver-promoted cyclization method.	*Tri*-O-acethl-D-glucal	22	[69]
Inoue et al.	2023	(1) Intermolecular and intramolecular radical coupling processes to link and cyclize the A- and C-ring fragments, respectively; (2) efficient decoration of the A- and C-ring functional groups using newly discovered chemo-, regio- and stereoselective processes; (3) finally, D ring formation and conjugation with amino-acid-delivered taxol.	2,2-Dimethylcyclohexane-1,3-dione	34	[70]

**Table 4 molecules-28-07517-t004:** Endophytic fungi producing paclitaxel from different hosts and their yields.

Family	Fungus	Host	Strain	Yield (μg/L)	Reference
*Taxaceae*	*Alternaria alternata*	*T. hicksii*	Tbp-9	0.13	[92]
	*Alternaria alternata*	*T. hicksii*	-	332–512	[89]
	*Alternaria alternata*	*T. chinensis* var *mairei*	TPF6	84.5	[93]
	*Alternaria* sp.	*T. cuspidata*	Ja-69	0.16	[94]
	*Alternaria alternata*	*T. cuspidata*	F3	195.4	[95]
	*Anthina Fr.*	*T. yunnanensis*	Tax-15	6.23	[96]
	*Aspergillus candidus*	*Taxus media*	MD2	112	[97]
	*Aspergillus candidus*	*T. media*	MD3	73	[98]
	*Aspergillus fumigatus*	*Taxus* sp.	TPF-06	1590.00	[99]
	*Aspergillus niger*	*T. cuspidata*	HD86-9	273.46	[100]
	*Aspergillus niger*	*Taxus yunnanensis*	IBFC-Z3S	1000	[101]
	*Aspergillus niger* var *taxi*	*T. cuspidata*	-	91	[102]
	*Bionectria* sp.	*T. chinensis* var *mairei*	XH004	33.90–430.46	[103]
	*Botryodiplodia theobromae*	*T. baccata*	BT115	280.5	[104]
	*Botrytis* sp.	*T. cuspidata*	HD181-23	206.34	[105]
	*Botrytis* sp.	*T. chinensis* var *mairei*	XT-2	161.24	[106]
	*Botrytis taxi*	*T. cuspidata*	HD104	-	[107]
	*Cephalosporium* sp.	*T. yunnanensis*	Tax-36	3.781	[96]
	*Chaetomium* sp.	*T. yunnanensis*	Tax-60	21.1	[96]
	*Cladosporium cladosporioides*	*T. media*	MD2	80	[97]
	*Didymostilbe* sp.	*T. chinensis* var *mairei*	DF110	-	[108]
	*Dimemasporium* sp.	*T. yunnanensis*	Tax-35	3.34	[96]
	*Ectostroma* sp.	*T. chinensis* var *mairei*	XT 5	276.75	[106]
	*Ectostroma* sp.	*T. yunnanensis*	Tax-16	4.092	[96]
	*Ectostroma* sp.	*T. yunnanensis*	Tax-25	2.16	[96]
	*Fusarium anthrosporioides*	*T. cuspidata*	F-40	131	[109]
	*Fusarium lateritium*	*T. baccata*	Tbp-9	0.13	[94]
	*Fusarium mairei*	*T. chinensis*	-	78	[110]
	*Fusarium mairei*	*Taxus* × *media*	UH23	20	[111]
	*Fusarium redolens*	*T. baccata*	TBPJ-B	66	[112]
	*Fusarium solani*	*T. chinensis*	Tax-3	164	[113]
	*Fusarium* sp.	*T. chinensis* var *mairei*	D62	148.95	[114]
	*Fusarium* sp.	*T. chinensis* var *mairei*	Y1117	2.70	[115]
	*Gliocladium* sp.	*T. baccata*	-	90	[110]
	*Gonatobotrys* sp.	*T. yunnanensis*	Tax-13	4.092	[96]
	*Guignardia mangiferae*	*Taxus* × *media*	HAA 11, HBA 29	-	[116]
	*Hypocrea* sp.	*T. media*	Z58	2.50–3.00	[117]
	*Metarhizium anisopliae*	*T. chinensis*	H-27	846.10	[118]
	*Monochaetia* sp.	*T. baccata*	Tbp-2	0.1	[92]
	*Mucor rouxianus*	*T. chinensis*	DA10	30	[119]
	*Mucor* sp.	*T. yunnanensis*	Tax-56	1.08	[96]
	*Mucor* sp.	*T. media*	060B1	2.50–3.00	[120]
	*Nodulisporium sylviforme*	*T. cuspidata*	HQD33, HQD48	51.06–125.70	[121]
	*Nodulisporium sylviforme*	*T. cuspidata*	NCEU-1	314	[121]
	*Nodulisporium sylviforme*	*T. cuspidata*	UV40-19, UL50-6	392	[121]
	*Nodulisporium sylviforme*	*T. cuspidata*	HDF68	468.62	[122]
	*Nodulisporium sylviforme*	*T. cuspidata*	-	450	[89]
	*Nodulisporium sylviforme*	*T. cuspidata*	HDFS_4-26_	516.37	[105]
	*Ozonium* sp.	*T. chinensis* var *mairei*	BT 2	4–18	[123]
	*Papulaspora* sp.	*T. chinensis* var *mairei*	XT17	10.25	[106]
	*Penicillium* sp.	*T. yunnanensis*	Tax-20	8.24	[96]
	*Pestalotia bicilia*	*T. baccata*	Tbx-2	1.08	[92]
	*Pestalotiopsis microspora*	*T. walichiana*	Ne-32	50.00	[92]
	*Pestalotiopsis microspora*	*T. cuspidata*	Ja-73	0.27	[92]
	*Pestalotiopsis* sp.	*T. yunnanensis*	YN6	120–140	[124]
	*Pestalotiopsis terminaliae*	*T. arjuna*	TAP 15	211.10	[125]
	*Phoma* sp.	*T. yunnanensis*	Tax-26	18.56	[96]
	*Phoma* sp.	*T. yunnanensis*	Tax-47	47.302	[96]
	*Phomopsis* sp.	*T. cuspidata*	BKH27	418	[126]
	*Pithomyces* sp.	*T. sumatrana*	P-96	0.095	[92]
	*Placodium* sp.	*T. yunnanensis*	Tax-24	13.63	[96]
	*Placodium* sp.	*T. yunnanensis*	Tax-49	31.06	[96]
	*Placodium* sp.	*T. yunnanensis*	Tax-55	0.46	[96]
	*Placodium* sp.	*T. yunnanensis*	Tax-63	3.11	[96]
	*Placodium* sp.	*T. yunnanensis*	Tax-65	6.27	[96]
	*Rhizoctonia* sp.	*T. yunnanensis*	Tax-1	1.43	[96]
	*Rhizopus*	*T. media*	M57	45.00–50.00	[127]
	*Stemphylium sedicola*	*T. baccata*	SBU-16	6.90	[128]
	*Taxomyces andreanae*	*T. brevifolia*	Tbp-2	0.02–0.05	[129]
	*Trichoderma* sp.	*T. yunnanensis*	Tax-23	19.59	[96]
	*Tubercularia* sp.	*T. chinensis* var *mairei*	TF-5	185.40	[130]
*Rhizosphere*	*Alternaria* sp.	*Rhizosphere*	-	4.2	[110]
	*Aspergillus flavipes*	*Rhizosphere*	-	185–850	[110]
	*Aspergillus flavus*	*Rhizosphere*	-	2.8	[110]
	*Aspergillus oryzae*	*Rhizosphere*	-	3.2	[110]
	*Penicillium chrysogenum*	*Rhizosphere*	-	85	[110]
	*Pestalotiopsis malicola*	*Rhizosphere*	-	186	[131]
*Bromeliaceae*	*Fusarium proliferatum*	*Tillandsia usneoides*	-	165	[110]
	*Pestalotiopsis humus* 133	*Tillandsia usneoides*	-	6.1	[110]
	*Pestalotiopsis humus* 154	*Tillandsia usneoides*	-	5.7	[110]
	*Pestalotiopsis* sp. 118	*Tillandsia usneoides*	-	8.9	[110]
	*Pestalotiopsis* sp. 107	*Tillandsia usneoides*	-	89	[110]
	*Pestalotiopsis* sp. 155	*Tillandsia usneoides*	-	4.3	[110]
	*Pestalotiopsis* sp. 163	*Tillandsia usneoides*	-	4.0	[110]
	*Phomopsis* sp. 116	*Tillandsia usneoides*	-	22	[110]
*Araucariaceae*	*Pestalotiopsis guepinii*	*Wollemia nobilis*	w-1, f-2	0.49	[132]
	*Pestalotiopsis* sp.	*Wollemia nobilis*	w-x-3	0.13	[132]
	*Pestalotiopsis* sp.	*Wollemia nobilis*	w-1, f-1	0.17	[132]
	*Phomopsis* sp.	*Wollemia nobilis*	-	170	[129]
*Cupressaceae*	*Fusarium mairei*	*Taxodium distichum*	UH23	20.00	[111]
	*Pestalotiopsis microspora*	*Taxodium distichum*	Cp-4	0.01–1.49	[133]
*Rutaceae*	*Bartalinia robillardoides*	*Aegle mamelos*	-	187.6	[134]
	*Phyllosticta citricarpa*	*Citrus media*	-	265.00	[110]
*Ginkgoaceae*	*Phoma betae*	*Ginkgo biloba*	SBU-16	795.00	[135]
	*Phomopsis* sp.	*Ginkgo biloba*	-	372	[136]
*Rubiaceae Juss.*	*Lasiodiplodia theobromae*	*Morinda citrifolia*	-	120	[110]
	*Pestalotiopsis microspora*	*Maguireothamnus speciosus*	-	0.11	[92]
*Podocarpaceae*	*Aspergillus fumigatus*	*Podocarpus* sp.	EPTP-1	560.00	[137]
*Sapindaceae Juss.*	*Pestalotiopsis pauciseta*	*Cardiospermum helicacabum*	CHP-11	113.30	[138]
*Combretaceae R. Br.*	*Pestalotiopsis terminaliae*	*Terminalia arjuna*	TAP-15	211.10	[139]
*Apocynaceae Juss.*	*Phyllosticta tabernaemontanae*	*Wrightia tinctoria*	-	461.00	[140]
*Sterculiaceae*	*Phyllostica melochiae*	*Melochia corchorifolia*	-	478	[110]
*Malvaceae Juss.*	*Phyllosticta dioscorea*	*Hibiscus rosa-sinensis*	-	298	[136]
*Moringa Adans.*	*Cladosporium oxysporum*	*Moringa oleifera*	-	550	[141]
*Betulaceae Gray*	*Penicillium aurantiogriseum*	*Corylus avellana*	NRRL 62431	70	[134]

**Table 5 molecules-28-07517-t005:** Heterologous production of taxane metabolites in different platforms.

Products	Concentration	Host	Reference
Taxadiene	1.0 g/L	*E. coli*	[158]
Oxygenated taxanes	570 mg/L	*E. coli*	[159]
Taxadiene	1.3 mg/L	*E. coli*	[160]
Oxygenated taxanes	33 mg/L	*E. coli* and *S. cerevisiae*	[161]
Taxadiene	1.98 mg/L	*Bacillus subtilis*	[162]
Taxadiene and taxadiene-5α-ol	1.0 mg/L and~25 μg/L	*S. cerevisiae*	[163]
Taxadiene	8.7 mg/L	*S. cerevisiae*	[164]
Taxadiene	72.8 mg/L	*S. cerevisiae*	[165]
Taxadiene	20 mg/L	*S. cerevisiae*	[166]
Taxadiene	129 mg/L	*S. cerevisiae*	[167]
Taxadien-5α-yl-acetate and total oxygenated taxane	3.7 mg/L and 78 mg/L	*S. cerevisiae*	[168]
Taxadiene and taxadien-5α-yl-acetate	71 mg/L and 21 mg/L	*S. cerevisiae*	[169]
Taxadiene	600 ng/g DW	*A. thaliana*	[170]
Taxadiene	160 mg/kg	Tomato fruits	[171]
Taxadiene and 5(12)-oxa-(11)-cyclotaxane	no yield	Tobacco (*Nicotiana sylvestris*)	[172]
Taxadiene	27 μg/g DW	Tobacco (*Nicotiana benthamiana*)	[173]
Taxadiene and taxadiene-5α-ol	56.6 μg/g and 1.3 μg/g FW	Tobacco (*Nicotiana* *benthamiana*)	[174]
Taxadiene	87.7 μg/g DW	*Nicotiana tabacum* cv. Petit Havana	[175]
TS-transgenic ginseng	14.6–15.9 μg/g DW	Ginseng (*Panax ginseng*) roots	[176]
Taxadiene	0.05% FW of plant tissue	*Physcomitrella patens* (moss)	[177]
Taxadiene	61.9 μg/L	*Alternaria alternata* (endophytic fungus)	[178]

**Table 6 molecules-28-07517-t006:** The production methods of paclitaxel and their advantages and disadvantages.

Methods	Advantages	Disadvantages
Extraction from plants	- original production process - main source	- long maturity time of the trees - extraction using organic solvents
Total synthesis	- obtained many paclitaxel analogues - organic synthesis of paclitaxel has been greatly enriched	- academic-level pursuits - large-scale production is extremely unlikely
Semi-synthesis	- rich raw materials - high yield - suitable for industrialized production	- high cost - relatively complicated synthesis process
Tissue and cell culture	- alleviate the dependence on the plant - avoids the introduction of exogenous genes that produce cytotoxicity - no transgenic manipulations	- instability of cell lines in the long fermentation periods - poor yields in the fermenters, even with elicitors
endophytic fungi method	- simple medium composition - fast growth rate - controllable conditions and low cost	- fungal storage - the decay and loss of productivity caused by multiple passaging cultures
Synthetic biology method	- easy operation and precise cellular modification - avoids unnecessary cellular metabolites	- heterologous synthesis of paclitaxel stops at the first 2–3 steps - microbial semi-synthesis still relies upon plant material

## Data Availability

All the data are included in the paper.

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
