# Peer review of "Research Advances in Clinical Applications, Anticancer Mechanism, Total Chemical Synthesis, Semi-Synthesis and Biosynthesis of Paclitaxel"

_molecules, 2023, doi:10.3390/molecules28227517_

Round 1

Reviewer 1 Report

Comments and Suggestions for Authors

The article entitled "Research advances in clinical applications, anticancer mechanism, total chemical synthesis, semi-synthesis and biosynthesis of paclitaxel" provides a comprehensive overview of recent advances in the production and application of paclitaxel.

The authors examined various methods, such as chemical synthesis, artificial culture, microbial fermentation and tissue cell culture, for obtaining paclitaxel to meet the clinical demand for this drug. The article also highlights the low toxicity, high potency and broad-spectrum anti-cancer activity of paclitaxel, making it one of the most effective natural anti-cancer drugs.

The authors have provided a theoretical basis and reference for future research into the production and application of paclitaxel. Overall, the manuscript is a well-written and organised, therefore I recommend its publication in its current form.

Reviewer 2 Report

Comments and Suggestions for Authors

The manuscript is highly interesting It fits with the scope of Molecules, and is recommended for publication after a minor revision.

Pagination is wrong.

Et al. should be in italic throughout the manuscript.

Line 27: Rewrite last part of the sentence (..with many 27 useful drugs are derived from plants)

Line 155: Use paclitaxel rather than taxol

Line 160: What is meant by “three formal syntheses”?

Line 172: Yield is low! What range?

Figure 4 is nice! Shouldn’t there be any references included in the text here as well in figs 5-9+11?

Line 183: Write full name of BMS first time it is mentioned in the text (Bristol Myers Squibb).

Line 194: Remove ‘company’

Table 4: Remove last column (Detection method). Analysis is not discussed in the text, and it is thus meaningless to include this information here.

Line 272: Remove: ‘…by HPLC-MS.’

Line 289: Remove: ‘…which was quantitatively analyzed by HPLC.’

Figure 10: The names of the two last molecular structures use taxol. It should rather be paclitaxel.

Line 393: Rather: ‘In sum, paclitaxel has been…’

Conclusion section is too long. Reduce.

Reviewer 3 Report

Comments and Suggestions for Authors

 The presented manuscript is an excellent overview of modern methods for obtaining paclitaxel, including its isolation from plant materials, total chemical synthesis, semi-synthesis and biosynthesis, as well as research advances in its clinical applications and anti-cancer mechanism.

However, I have a few comments on the text of the manuscript.

In the text of the manuscript, the names of the authors of cited sources are indicated in different manners - Wani (line 34), Dr. Susan B. Horwitz (line 42), Pierre Potier (line 94), Min et al. (line 147). Apparently, this happens because individual parts of the manuscript were written by different authors, however, this must be unified.

In some places there is a noticeable lack of references, for example – “Anticancer activity of paclitaxel was demonstrated in the mouse melanoma B16 model in 1976” (line 41) or “It has been clinically proven that paclitaxel has good anti-tumor effects, especially for ovarian cancer, uterine cancer and breast cancer, which have a high incidence of cancer.” (lines 47-48), “In 1996, docetaxel was marketed for 96 the treatment of breast, colon and NSCLC.” (lines 96-97).

 In 1994, Nicolaou et al. and Holton et al. reported the first total synthesis of paclitaxel, and subsequently, various total synthesis methods were reported [44,45].” (lines 157-158) – it seems that these references should appear in the next sentence – “So far, more than 60 research groups around the world have already culminated in eleven total syntheses and 159 three formal syntheses, as well as more than 60 synthetic model studies of paclitaxel.”, whereas Ref. 46 and 47,48 should be here.

Section 3.1. is well written, however the number of references here is extremely insufficient. The next references should be added among others - J. Nat. Prod., 1999, 62, 2, 244–247; Biotechnol. Progr., 2012, 28, 4, 990-997; Molecules, 2017, 22, 9, 1483; Korean J. Chem. Eng., 2022, 39, 3389–3398; Biotechnol. Bioproc. Eng., 2023, 28, 336–344; Biotechnol. Bioproc. Eng., 2023, 28, 545–553;

Line 51. “10 000 kg” is better than “ten thousand kg”.

Line 74. Probably “DNA molecules” and not “DAN molecules”.

Lines 147-148. “negative pressure” is not good term, “vacuum“ is better.

In Table 3, some of the source materials are given precisely and some are given very generally. If possible, authors should try to unify this.

Unfortunately, I was unable to find references to works 83-96 in the text of the manuscript. I suggest that they should be in Table. 4. The authors should fix this.

Line 404. Not “America”, but “United States”.

In some references, the years of publication are indicated along with the letters (2017a, 2017b, etc.) - it seems that the manuscript was originally prepared for a journal of a different format. This needs to be fixed.
